# Emergency FMD Serotype O Vaccines Protect Cattle against Heterologous Challenge with a Variant Foot-and-Mouth Disease Virus from the O/ME-SA/Ind2001 Lineage

**DOI:** 10.3390/vaccines9101110

**Published:** 2021-09-29

**Authors:** Nagendrakumar Balasubramanian Singanallur, Aldo Dekker, Phaedra Lydia Eblé, Froukje van Hemert-Kluitenberg, Klaas Weerdmeester, Jacquelyn J Horsington, Wilna Vosloo

**Affiliations:** 1Australian Centre for Disease Preparedness, CSIRO-Health & Biosecurity, 5 Portarlington Road, Geelong, VIC 3220, Australia; jacquelynjh@gmail.com (J.J.H.); wilna.vosloo@csiro.au (W.V.); 2Laboratory Vesicular Diseases, Department of Virology and Molecular Biology, Wageningen Bioveterinary Research, Houtribweg 39, 8221 RA Lelystad, The Netherlands; Aldo.Dekker@wur.nl (A.D.); phaedra.eble@wur.nl (P.L.E.); W.Hemert@upcmail.nl (F.v.H.-K.); KWeerd@hotmail.com (K.W.)

**Keywords:** foot-and-mouth disease virus, vaccine efficacy, serotype O/ME-SA/Ind2001 variant, heterologous challenge, cattle

## Abstract

Vaccination is one of the best approaches to control and eradicate foot-and-mouth disease (FMD). To achieve this goal, vaccines with inactivated FMD virus antigen in suitable adjuvants are being used in addition to other control measures. However, only a limited number of vaccine strains are commercially available, which often have a restricted spectrum of activity against the different FMD virus strains in circulation. As a result, when new strains emerge, it is important to measure the efficacy of the current vaccine strains against these new variants. This is important for countries where FMD is endemic but also for countries that hold an FMD vaccine bank, to ensure they are prepared for emergency vaccination. The emergence and spread of the O/ME-SA/Ind-2001 lineage of viruses posed a serious threat to countries with OIE-endorsed FMD control plans who had not reported FMD for many years. In vitro vaccine-matching results showed a poor match (r_1_-value < 0.3) with the more widely used vaccine strain O_1_ Manisa and less protection in a challenge test. This paper describes the use of the O3039 vaccine strain as an alternative, either alone or in combination with the O_1_ Manisa vaccine strain with virulent challenge by a O/ME-SA/Ind-2001d sub-lineage virus from Algeria (O/ALG/3/2014). The experiment included challenge at 7 days post-vaccination (to study protection and emergency use) and 21 days post-vaccination (as in standard potency studies). The results indicated that the O3039 vaccine strain alone, as well as the combination with O_1_ Manisa, is effective against this strain of the O/ME-SA/Ind/2001d lineage, offering protection from clinical disease even after 7 days post-vaccination with a reduction in viraemia and virus excretion.

## 1. Introduction

Foot-and-mouth disease (FMD) is a contagious infectious disease of cloven-hoofed animals. The disease presents as lesions on areas of friction such as the mouth, feet, and teats in lactating animals, but sub-clinical infections can also occur, especially in sheep and goats. The disease is caused by the FMD virus (FMDV), belonging to the *Aphthovirus* genus of the family *Picornaviridae*. Seven serotypes of FMDV (A, O, C, Asia 1, and South African Territories 1, 2, and 3) have been identified, and multiple genotypes/topotypes and lineages occur within each serotype [1,2].

FMD is widely prevalent, with the disease circulating in regions estimated to contain 77% of the global livestock population, affecting a large proportion of animals during outbreaks, often involving multiple species at the same time. These factors collectively lead to a huge disease burden and have a significant global animal health and socio-economic impact [3]. The disease can potentially have serious economic and social implications if infection occurs in countries that are FMD-free and are major producers and exporters of livestock and livestock products. For example, most of the economic costs of a hypothetical FMD outbreak in Australia arise from revenue losses caused by immediate and prolonged import bans by Australia’s main trading partners, which are estimated to be in excess of $50 billion over 10 years [4]. Maintenance of the World Organisation for Animal Health (OIE) FMDV-free status is critical for the free trade of animals and animal products.

One of the main approaches to FMD control and eradication in endemic countries is through vaccination with inactivated FMDV antigen formulations with suitable adjuvants [5]. Vaccination was successfully used to control the disease in several formerly endemic countries in Europe and most countries in South America [6]. In addition, the unprecedented magnitude of slaughter to eradicate FMDV in the UK underlined the need for alternative policies to ‘stamping out’ in FMD-free countries where vaccination is not routinely used [7]. Since this incursion into the European Union, most countries free from FMD will now consider emergency vaccination, based on their own risk assessment, should an outbreak occur [8].

During 2013, a new sub-lineage of serotype O FMDV, O/ME-SA/Ind2001, previously restricted to South Asia, emerged and spread within the Middle East and North Africa [9]. Different variants of the same virus were identified in the Middle East in the past, causing sporadic outbreaks in Kuwait (1997), Oman (2001) and UAE (2008) [10]. Algeria, Tunisia, and Morocco in North Africa, which had OIE-endorsed FMD control plans and had not reported FMD for many years, witnessed widespread outbreaks due to the O/ME-SA/Ind2001d lineage during 2014 [11]. Similar viruses were also reported from South East Asian nations: Lao PDR, Myanmar and Vietnam (2015) and Thailand (2016) [10,12]. Several sub-lineages (a–e) have appeared to date, with the recent emerging sub-lineage e moving into Pakistan [13,14]. The antigen-matching studies carried out by the World Reference Laboratory (WRL) for FMD showed that these isolates had a poor antigenic match with the O_1_ Manisa vaccine strain, but a moderate to good match with the O3039 vaccine strain [9].

A previous study in cattle using a high-potency O_1_ Manisa vaccine (estimated 17 PD_50_/dose) and challenge with an O/ME-SA/Ind2001d lineage virus isolated from Algeria (O/ALG/3/2014) resulted in a heterologous PD_50_ value of approximately 3.5 [11]. The challenge virus (O/ALG/3/2014) belongs to the O/ME-SA/Ind2001d lineage and is heterologous to both the O_1_ Manisa and O3039 vaccine strains with relative homology r_1_-value < 0.3 by virus neutralisation test [9]. The study concluded that the high-potency O_1_ Manisa vaccine would be effective against the O/ME-SA/Ind2001 lineage of viruses, but further studies to evaluate other vaccine candidates or multivalent combinations should be carried out for potential emergency purposes in FMD-free settings [11]. The emergence of O/ME-SA/Ind2001 lineage and subsequent spread into South East Asia poses a risk to the Australian livestock industry and to other FMD-free countries in the region. Its spread westward has caused severe economic losses and increased the disease burden in Northern Africa and Middle East Asia. Therefore, the aim of the present study was to test the efficacy of an O3039 vaccine alone and in combination with O_1_ Manisa (both formulated as emergency vaccine) in cattle at different timepoints post-vaccination when challenged with an O/ME-SA/Ind2001 lineage virus, and make recommendations on the vaccine strains to be included in vaccine banks.

## 2. Materials and Methods

### 2.1. Experimental Animals

The vaccine efficacy study was performed in the animal facility of Wageningen Bioveterinary Research (WBVR), Lelystad, The Netherlands, according to protocols for experimentation with live cattle approved by the Animal Ethics Committee of the Australian Animal Health Laboratory (AEC 1754) and the Dutch Animal Ethics Law (2016003/LVZ194). In total, 23 Holstein Friesian or Holstein Friesian cross-bred cattle (aged 8–12 months and weighing approximately 200–250 kg) were used for the study. The cattle were housed in a tie-stall to limit the transmission of virus between the cattle and ensure independent observations per cow.

### 2.2. Viruses and Vaccines

FMDV O/ALG/3/2014 (O-ALG) cattle passaged challenge virus was obtained from the Pirbright Institute [11], titrated on primary lamb kidney cells and was found to have a titre of 7.95 log_10_ plaque-forming units (PFU)/mL. The virus was originally isolated by the Istituto Zooprofilattico Sperimentale della Lombardia dell’Emilia Romagna, Italy, from an outbreak in Algerian cattle during 2014, and subsequently passaged in cattle to prepare the cattle challenge virus, and in cell culture for in vitro assays. The challenge virus was diluted in Minimum Essential Medium with Hanks’ balanced salts, 2% foetal bovine serum (FBS) and antibiotics (final concentration 20 IU/mL penicillin, 20 mg/mL streptomycin, 0.4 mg/mL amphotericin B, 10 mg/mL polymyxin B and 48 mg/mL kanamycin sulphate) to give a final titre of 5.7 log_10_ PFU/mL. O3039 and O_1_ Manisa viruses for virus neutralisation test (VNT) were supplied by WBVR, Lelystad. High-potency double oil emulsion (DOE) vaccines, formulated with proprietary adjuvants, with a combination of vaccine strains O3039 and O_1_ Manisa (Combo; both strains at >6 PD_50_/dose) or O3039 alone (>6 PD_50_/dose), were prepared from the antigen reserve of the Australian FMD vaccine bank by Merial Company Limited, United Kingdom, and 2 mL was administered intramuscularly in the side of the neck, according to the manufacturer’s recommendations.

### 2.3. Immunisation, Challenge, Clinical Score and Sample Collection

Four groups of 5 cattle were vaccinated with either the bivalent Combo or a monovalent O3039 vaccine and challenged at 21 days post-vaccination (dpv, Combo-21 or O3039-21) or 7 dpv (Combo-7 or O3039-7). The vaccinated cattle and three unvaccinated control cattle (UVC) were challenged with 0.2 mL of 10^5.7^ PFU/mL FMDV O/ALG by intra-dermal inoculation (IDL) into two locations of the tongue, with 0.1 mL per location. For this study, only lesions on the feet were considered generalisation of disease and, therefore, not protected [15]. Mouth and nose lesions at sites other than the injection sites (dental pad, lips, etc.) were recorded but were not considered unprotected. Rectal temperatures were recorded daily, and animals were examined under anaesthesia for clinical lesions on 4 and 8 days post-challenge (dpc). Pain relief was pre-emptively given 0–4 days after challenge, and then at the onset of moderate clinical signs using Flunixin meglumine @3.33 mg/Kg body weight, equivalent to 1 mL/15 Kg body weight (Finadyne pour-on 50 mg/mL, MSD Animal Health, the Netherlands). The number of animals showing disease generalisation post challenge based on foot lesions were counted for each group and used to calculate the difference in protection between different groups. Clotted blood from Combo-21 or O3039-21 groups were sampled on −21, −18, −14, −10, −7, −4, 0, 1–7, 11, 14, 18, 21, 25, 28 and 32 dpc and the animals, while Combo-7 or O3039-7 groups were sampled on −7, −4, 0, 1–7, 11, 14, 18, 21, 25, 28 and 32 dpc. The unvaccinated control animals were sampled on 0, 1–7, 11, 14, 18, 21, 25, 28 and 32 dpc. The sera were collected after centrifugation and stored at −80° until testing. The oral fluid was collected on 0, 1–7, 11, 14, 18, 21, 25, 28 and 32 dpc with Salivette swabs, extracted using 0.5 mL of minimal essential medium supplemented with 5% fetal bovine serum and 10% antibiotic cocktail and transferred to fresh tubes. Nasal swabs were collected on the same days as the oral swabs using sterile cotton swabs and placed in 2 mL of minimal essential medium supplemented with 5% fetal bovine serum and 10% antibiotic cocktail [16,17,18,19,20]. Oro-pharyngeal fluids were collected using probang cups on 0, 11, 14, 18, 21, 25, 28 and 32 dpc and diluted 1:1 in Earle’s MEM (2% fetal bovine serum and 2% antibiotic cocktail [21]. All samples were stored at −80° until testing.

### 2.4. Serological Assays to Measure Antibodies

Serum samples were examined for FMDV neutralising antibodies using VNT, performed using IBRS-2 cells following standard procedures [15,22]. The VNT was performed to assess the neutralising antibody titre against the homologous vaccine strains O3039 and O_1_ Manisa, and the heterologous challenge virus O-ALG. The neutralising antibody titres were calculated as the log_10_ of the reciprocal of the final antibody dilution required for 50% neutralisation of 100 TCID_50_ of virus. Animals that showed a titre of log_10_ > 1.20 were considered positives [15]. Antibodies to the non-structural proteins (NSP) of FMDV were detected using the PrioCHECK^®^ FMDV-NS kit (Thermo Fisher, Waltham, MA, USA) and sera showing a percentage inhibition value (PI) of ≥50 were considered positives, according to the manufacturer’s instructions.

### 2.5. Viraemia and Virus Excretion

Viraemia (serum) and virus excretion in oral and nasal swabs and probang samples were studied by virus titration and real-time RT-PCR. Primary ovine kidney cells supplied by the WBVR were used to perform virus isolation and titration using a standard plaque assay, and the results expressed as log_10_ PFU/mL [23]. Samples were recorded as negative when no plaques were observed. As the plates were washed and stained, a second passage was not possible. Total RNA from 200 µL of serum, oral and nasal secretions (as mentioned above) and probang samples were isolated using the MagNA Pure 96 DNA and Viral NA Large Volume kit on the MagNA Pure 96 system (Roche^®^ Life Science) and real-time RT-PCR was carried out using the LightCycler RNA Amplification Kit Hybridisation Probes and LightCycler 480 (Roche^®^ Life Science) using the protocol described by Moonen et al. 2003 [24]. Samples were declared positive when the fluorescence signal rose above the background signal (crossing point determined automatically by the second derivative maximum method for quantification by the software supplied by Roche^®^).

### 2.6. Statistical Analysis

Clinical protection based on count data were analysed using the two-sided Fischer exact test. Group means and standard deviations were calculated and expressed as Mean ± SD. Longitudinal data for continuous outcomes in multiple vaccine groups were analysed using a linear mixed-effects model (lme library) [25]. ANOVA was used to test the statistical differences between groups. If a statistical difference was found, a pairwise *t*-test (with Holm correction) was used to analyse differences between groups. Longitudinal data (virus isolation, RT-PCR results and NSP response) were analysed using the animal number as random variable and dpc, group and vaccination (yes or no) as possible explanatory variables. Using forward selection, the best model, with the lowest Akaike’s Information Criterion (AIC), was chosen. For the NSP responses, the data from 0 to 35 dpc were analysed, with the percent-inhibition value (PI) used as a response variable. Cattle number was added as a random variable. Dpc (as a factor) and vaccine group were analysed as explanatory variables, as well as interactions. Data on virus isolation and RT-PCR were also analysed the same way. In all models, explanatory variables were selected based on the lowest AIC using forward selection. A statistical analysis was performed using R version 4.0.2 [26]. All results are presented in the Supplementary Word file.

## 3. Results

### 3.1. Post Challenge Outcomes (Temperature and Clinical Disease)

The clinical outcome of the vaccine efficacy studies is summarised in Appendix A. All cattle challenged at 21 dpv (Combo-21 and O3039-21) had no lesions on their feet and were considered protected. The three UVC cattle, as well as one in the Combo-7 (#9672) and two in the O3039-7 vaccinated groups (#9674 and 9677), showed generalization, with lesions on one or more feet. The difference in clinical outcome between the four vaccine groups was not significant (*p* = 0.561; Fisher’s Exact Test, two-sided, Appendix A), but when compared with the unvaccinated control groups, there was a significant difference (*p* = 0.01129; Fisher’s Exact Test, two-sided, Appendix A). The difference in rectal temperatures between the vaccine groups was statistically insignificant (*p* > 0.05) but was significant when compared with the control group (*p* = 0.045). However, there was a significant difference in rectal temp between 0 dpc and 1–4 dpc in all groups (1–3 dpc—*p* < 0.001 and 4 dpc—*p* = 0.0076)) (Figure 1; Appendix A).

### 3.2. Serological Response to Vaccination at the Time of Challenge

The sera from the four vaccinated and one unvaccinated control groups were all tested for neutralising antibodies using VNT against O3039, O_1_ Manisa and O-ALG. The control animals had no detectable neutralising antibodies to any of the three viruses on the day of challenge. All vaccinated animals developed antibody titres against their vaccine virus(es) but also against the other type O virus(es). In the vaccinated groups, higher titres were observed at 21 dpv against all three viruses compared to 7 dpv (Figure 2).

Comparison of the neutralising antibodies at the time of challenge in a linear mixed model showed a significant difference between the virus used in the VNT as well as a significant difference for the interval between vaccination and challenge, but no differences between the neutralising antibody titres induced by the bivalent (Combo) and monovalent (O 3039) vaccine. The mean O3039 and O_1_ Manisa response differed significantly between the 21 and 7 dpv groups (*p* = 0.004 and *p* = 0.0004, respectively). However, the mean O-ALG response did not differ significantly between the 21 and 7 dpv groups (*p* > 0.05) (Figure 2; Appendix A).

All animals responded to challenge with an increase in neutralising antibody and the differences in post-challenge response were not statistically significant (Appendix A).

### 3.3. NSP Antibody Responses

Antibodies to the NSP of FMDV developed at a timepoint between 5 and 11 dpc in all animals (sera were not collected daily). The linear mixed regression model analysis showed that dpc (as a factor), group and interaction between dpc and group best explained the NSP response (Supplementary File). The interaction shows that there was no significant difference (*p* > 0.05) between the vaccine groups, but a significant difference between day of challenge (21 or 7 dpv; *p* < 0.05) and post-challenge on 28 and 32 dpc (Figure 3; Appendix A).

### 3.4. Viraemia and Virus Excretion in Clinical Samples

Infectious FMD virus and viral RNA were detected in the sera of all UVC cattle at 1–3 dpc. Except for one vaccinated animal, where viral RNA could be detected at 1 dpc (O3039-7 group (#9677)), no virus or RNA was detected in the sera of the cattle in the vaccine groups (Figure 4).

The oral swabs from all groups collected between 1 and 7 dpc were positive for infectious virus and/or viral RNA due to the tongue lesions that occurred as a result of the IDL challenge and there was not a significant difference between the groups. Once the lesions were healed, virus was not detected in the oral swabs up to 32 dpc (Figure 4).

Viral RNA and/or virus was intermittently detected in nasal secretions of all groups regardless of vaccination status between 1 and 7 dpc. The number of positive detections decreased over time, with only two positive RNA samples at 7 dpc in vaccinated groups challenged 7 dpv. Significantly lower levels of infectious virus and/or viral RNA were detected in the nasal secretions of vaccinated animals when compared to the unvaccinated animals (one-way ANOVA—*p* < 0.001) per day or over the total amount excreted during the viraemic and clinical phase of the disease (1–7 dpc). The viral RNA levels in the Combo-21 and O3039-21 cattle differed significantly from the Combo-7 and O3039-7 cattle (one-way ANOVA—*p* < 0.05). (Figure 4).

### 3.5. Virus Persistence and Carrier Status

Virus and viral RNA were detected in the probang samples of most vaccinated animals from 11 dpc and there was no difference between the vaccinated and control groups. There was an increase in the number of positive animals in group Combo-21 up to 32 dpc (Figure 4). The infectious virus titres were low and never exceeded log_10_ 2 PFU/mL. The probang samples of only one of the unvaccinated control animals were positive between 18 and 32 dpc. These results indicated that 2–3 cattle in each of the vaccine groups and one in the UVC group developed virus persistence and could be considered carriers.

## 4. Discussion

The emergence of the novel O/ME-SA/Ind2001 lineage of FMD viruses continues to pose a major threat to the ‘FMD free’ countries in Europe and Northern Africa, and variants and several sublineages have emerged (a–e). An earlier study recommended the use of O_1_ Manisa vaccine strain for emergency use and control of outbreaks with O/ME-SA/Ind2001d sub-lineage and also suggested evaluation of other vaccine candidates (or multivalent combinations) that might potentially be used for emergency purposes in FMD-free setting [11]. The antigen-matching reports from the World Reference Laboratory in Pirbright indicated that an alternative vaccine strain, O3039, in the vaccine banks matched with the viruses of the above-mentioned lineage [9,27]. Keeping these results and recommendations in mind this study was designed to determine if the O3039 vaccine strain can provide better protection, either as a monovalent vaccine, or in combination with O_1_ Manisa. In addition, since these vaccines will be used during emergency responses in many countries, it was also necessary to determine how efficacious they will be soon after vaccination. In this study we compared the outcomes of virus challenge at 7 and 21 dpv. Both the O3039 monovalent vaccine and the combination vaccine with O_1_ Manisa (Combo) protected all cattle from clinical disease following challenge after 21 dpv with O/ALG/3/2014 (O-ALG). Showing early protection at only 7 days between vaccination and challenge, both vaccines protected, on average, 70% of the cattle. This is despite the severe challenge model with high doses of virus injected directly into the tongue. Under field conditions, vaccinated cattle will be challenged by natural routes and direct contact with infected animals, which is seen as less severe. Protection at early timepoints post-vaccination has been made in several studies [28,29,30], indicating the usefulness of emergency vaccines that often contain more antigen per dose than routine vaccines.

The O3039 monovalent vaccine and Combo vaccines were very effective in reducing or preventing viraemia against the O/ME-SA/Ind2001d lineage as early as 7 dpv, as evidenced by real-time RT-PCR that showed viraemia in only one of the animals challenged 7 dpv, whilst no viraemia was detected in any of the other animals and groups. This contrasted with the unvaccinated animals, where infectious virus or viral RNA was detected for 3 dpc. Viraemia usually coincides with the clinical phase of FMD; therefore, decreasing this phase is an important aspect of disease control by limiting the clinical signs and lowering the extent of virus shedding [31].

We found no correlation between pyrexia and the development of clinical signs, which confirms the previous findings that pyrexia is not a reliable predictor of the development of clinical FMD signs [11,30] and, therefore, not a practical indicator of FMDV infection. However, this could be a result of the administration of anti-pyrectics and analgesics (NSAIDs) for ethical considerations.

Virus/viral RNA was detected from the oral and nasal swabs between 1 and 7 dpc in all the groups, indicating that these samples are valuable for identifying infection during an outbreak for surveillance activities. There was a marked reduction in virus shedding in the nasal secretions of the vaccinated animals compared to the unvaccinated animals, which will result in less virus in the environment and assist with disease control, confirming findings from previous studies that showed that vaccination can help prevent the spread of the virus within the infected premises and between farms [32].

One important aspect for disease control is the prevention of persistent infection, the co-called ‘carrier state’, defined as the presence of virus/viral RNA 28 days post-infection [33]. Although the probability of transmission from persistently infected livestock to other susceptible animals is low [34,35,36] or absent [37], when providing evidence of freedom from infection for reasons such as accessing trade markets or changing country disease status, it is also important to demonstrate the absence of persistently infected animals. This often has economic impacts due to increased surveillance costs when using vaccination and is one of the deterrents to using vaccination to control outbreaks in previously free countries or regions. Previous studies found that approximately 50% of unvaccinated infected animals show virus persistence, while vaccination decreases the number of persistently infected animals [34,38] because the vaccinated animals did not show generalisation of the disease post-challenge. It was also shown that the number of persistently infected animals decreases over time [21]. In our study, viral RNA/virus was detected in most of the vaccinated animals, but at intermittent times and at very low titres (<2 PFU/mL) up to 32 dpc. Vaccination, therefore, did not prevent persistent infection at timepoints close to the challenge, but the level of live virus was low. However, despite this finding, disease models showed that there is no need to develop vaccines that provide sterile immunity [34,35] and that current vaccines are suitable for control campaigns.

Antigen-matching studies by national and international reference laboratories give an indication of the match between new emerging isolates and the current vaccine strains. These studies are generally performed with reference sera raised using commercial vaccine formulations (payload equivalent to 3–6 PD_50_/dose) and not emergency vaccine formulations (payload that provide > 6 PD_50_/dose). While r_1_-values provide valuable indications of vaccine efficacy when assessing homologous systems, the interpretation is less clear when the vaccine and virus are heterologous. In vivo challenge studies such as the one described here present valuable data on the efficacy of heterologous emergency vaccines.

## 5. Conclusions

The ever-evolving nature of FMDV poses a significant challenge to endemic and FMD-free countries that rely on vaccination to control the disease and incursions. Vaccine efficacy studies such as these provide additional evidence on whether a particular vaccine formulation will have a sufficient impact on clinical disease, virus excretion and persistent infection, even at early timepoints post-vaccination. These results are based on a small number of animals; therefore, it is prudent to expand such studies to more animals and other challenge viruses to confirm the efficacy of monovalent or combination vaccines. Given the epidemiological situation of FMD around the world and co-circulation of serotype O viruses belonging to different genotypes and lineages, we recommend that both O_1_ Manisa and O3039 are included in the vaccine banks.

## Figures and Tables

**Figure 1 vaccines-09-01110-f001:**
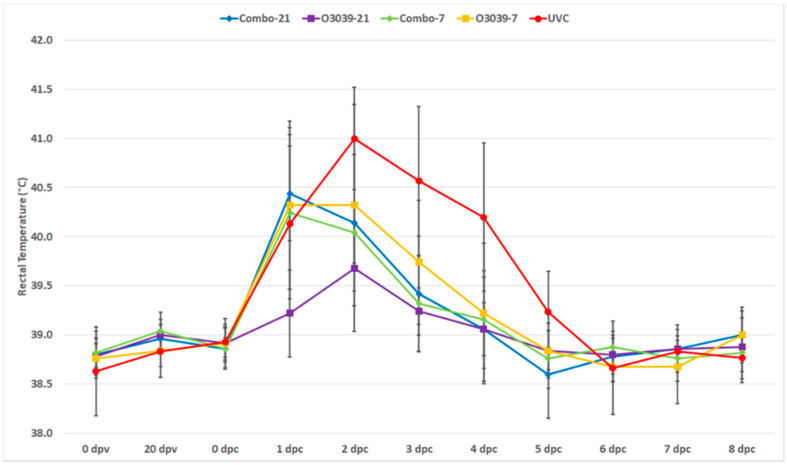
Mean rectal temperature (°C) of cattle given a full dose of a O3039 + O_1_ Manisa combination vaccine or O3039 monovalent vaccine and challenged on 21 or 7 days post vaccination, with O-ALG virus, with error bars representing standard error of the mean for each group. The broken line indicates temperature (39.5 °C) above which an animal is considered as having pyrexia; dpv—days post vaccination and dpc—days post challenge.

**Figure 2 vaccines-09-01110-f002:**
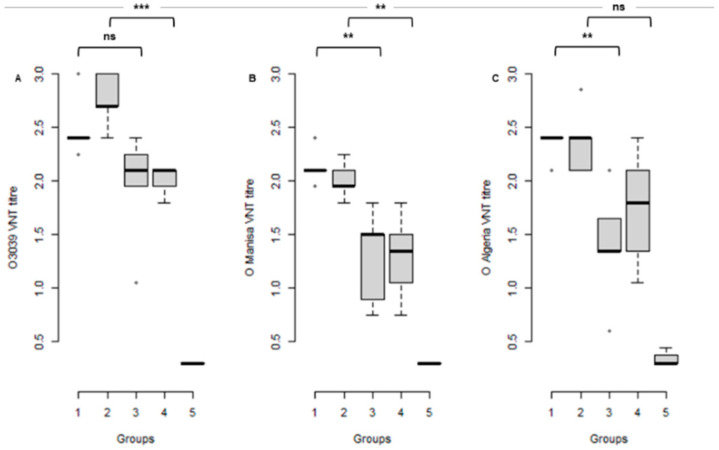
Box plot of serum antibody titres estimated by VNT of cattle vaccinated with O3039 + O_1_ Manisa combination vaccine (Combo) or O3039 monovalent vaccine on the day of challenge (21 or 7 days post vaccination) with O Algeria 2014 (O-ALG). The horizontal line represents the median titre for each group. The serum antibody titres were estimated against O3039 (**A**), O_1_ Manisa (**B**) and O-ALG (**C**) viruses. Key: 1 = Combo-21, 2 = O3039-21, 3 = Combo-7, 4 = O3039-7, 5 = Unvaccinated Control. ns—No significant difference; **—*p* < 0.01; ***—*p* < 0.001. The boxplots show the interquartile range (median represented as the thick horizontal line within the box, and the first and third quartile of the data) and the minimum and maximum values for each group connected to the boxes with the vertical line.

**Figure 3 vaccines-09-01110-f003:**
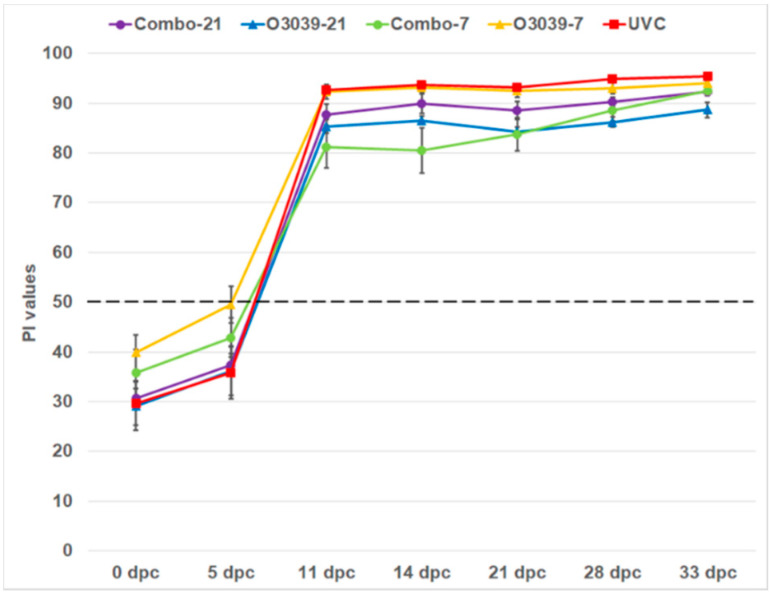
NSP antibody responses post-challenge in cattle given a O3039 + O_1_ Manisa combination vaccine or O3039 monovalent vaccine and in unvaccinated cattle. dpc—day post-challenge; PI—mean per cent inhibition; horizontal dashed line indicates the cut-off level; Error bars represent standard error of mean PI values.

**Figure 4 vaccines-09-01110-f004:**
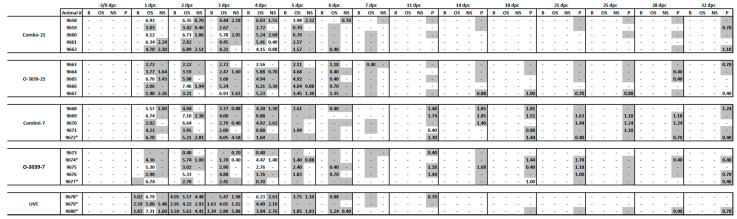
Virus isolation (log_10_ PFU per ml) and viral RNA detected in sera, swabs, and oro-pharyngeal fluids from 0 to 32 days post challenge. Animals were vaccinated with the O3039 + O_1_ Manisa combination vaccine or O3039 monovalent vaccine and challenged 21 or 7 days post-vaccination with O-ALG virus. dpc—day post-challenge; B—Blood, OS—Oral Swab, NS—Nasal Swab, P—Oro-pharyngeal fluids collected using a probing cup, Cells with grey shade are positive for FMDV genome by PCR, numbers indicate the viral titres and “-” Indicates below the limit of detection by PCR or VI.

## Data Availability

All data are archived as per the CSIRO policies and guidelines.

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
