# Peer review of "Emergency FMD Serotype O Vaccines Protect Cattle against Heterologous Challenge with a Variant Foot-and-Mouth Disease Virus from the O/ME-SA/Ind2001 Lineage"

_vaccines, 2021, doi:10.3390/vaccines9101110_

Round 1
Reviewer 1 Report
General comments:
- Please use subscript in case of PD50 (PD50) and superscript in case of titers (for example Lines 128 en 275).
- O Manisa or O1 Manisa?
Specific comments:
- In lines 89-91 is written “The emergence of O/ME-SE/Ind2001 (…) poses a risk to the Australian livestock industry.” It is not only a risk for the Australian livestock, it is a global issue. Please adjust the sentence accordingly.
- Line 118: Please provide the name of the adjuvant.
- In lines 130-131 is written: “For this study, only lesions on the feet were considered as generalization of disease and therefore vaccine failure.” Please use ‘not protected’ instead of ‘vaccine failure’ for 2 reasons: 1) the former term is used in the Results and Discussion section, and 2) an animal not protected does not mean that the vaccine has failed completely. Also, please refer to the relevant Ph. Eur/OIE guidelines to clarify why only lesions on the feet are considered as generalization of disease and lesions in the mouth are not considered.
- Line 137: Timepoints of blood sampling are described. Were blood samples also taken at -21, -18, -14 and -10dpc from animals that were challenged at 7dpv? Please clarify.
- Line 196: Was the difference of clinical outcome statistically significant between vaccinated groups and unvaccinated group?
- Figure 1: Please correct the legend, because the dashed lines cannot be discriminated from the normal lines. Figure 3 has a bit of the same problem.
- Figure 2: The boxplot is not clear. For some groups there is a grey box and for others not; for some groups there is a dashed line, while for others there are only diamonds visible. Shouldn't it all look the same? Please also explain in the figure's description how to read the figure.
- Figure 4: Small table, difficult to read. Please include averages per group.
- Line 254: replace O-AG with O-ALG
- Line 260: IDL challenge is not introduced earlier. It should probably be introduced in line 129.
- Lines 312-316: It is not clear whether anti-pyretics and analgesics were used in this study. If so, please include the relevant information in the M&M section.
- Lines 317-319: Is it novel to use swabs for FMD diagnostics? If not, please refer to studies in which this has been done before.
- Lines 344-350: The conclusion does not address the aim of this study (lines 91-96). Especially the phrase "and make recommendations on vaccine strains to be included into vaccine banks". Therefore, the authors are requested to provide their recommendation. Should only O1 Manisa, only O3039 or the combo by included in vaccine banks?
Author Response
Reviewer 1:
General comments:
- Please use subscript in case of PD50 (PD50) and superscript in case of titers (for example Lines 128 en 275).
Changed as PD50 and as TCID50 as recommended. To be consistent log10 is also changed to Log10
- O Manisa or O1 Manisa?
Both are interchangeable. Though the convention of using subtypes has been discontinued, the old vaccine strains still carry these subtype indications. To be consistent we have added the subtype and changed to O1 Manisa throughout the manuscript.
Specific comments:
- In lines 89-91 is written “The emergence of O/ME-SE/Ind2001 (…) poses a risk to the Australian livestock industry.” It is not only a risk for the Australian livestock, it is a global issue. Please adjust the sentence accordingly.
We specifically indicate that due to the closeness of South East Asia to Australia and the risk of incursion into Australia is from South East Asia. We take on board the suggestion and have reworded as follows:
The emergence of O/ME-SA/Ind2001 lineage and subsequent spread into South East Asia poses a risk to the Australian livestock industry and to other FMD free countries in the region. Its spread westward has caused severe economic losses and increased the disease burden in Northern Africa and Middle East Asia.
- Line 120: Please provide the name of the adjuvant.
The adjuvants are commercial secrets of vaccine manufacturing countries. The literature of the vaccine formulations say DOE vaccines and do not divulge the adjuvants.
Included in the text as “High potency double oil emulsion (DOE) vaccines formulated with proprietary adjuvants ….”
- In lines 130-131 is written: “For this study, only lesions on the feet were considered as generalization of disease and therefore vaccine failure.” Please use ‘not protected’ instead of ‘vaccine failure’ for 2 reasons: 1) the former term is used in the Results and Discussion section, and 2) an animal not protected does not mean that the vaccine has failed completely. Also, please refer to the relevant Ph. Eur/OIE guidelines to clarify why only lesions on the feet are considered as generalization of disease and lesions in the mouth are not considered.
Changed as follows in Lines 132-135: “Mouth and nose lesions at sites other than the injection sites (dental pad, lips, etc.) were recorded but not considered as not protected”.
With regards to the second part of the comment on mouth lesions, since the challenge is intradermolingual (IDL), we expect lesions in the mouth and protection is measured based on generalisation of disease and appearance of lesions on the foot. Only if the challenge methods are different, for example direct contact challenge with donor animals, intranasal instillation or aerosol challenge, mouth lesions are considered. The OIE FMD Manual 2017 is added as reference now.
- Line 137: Timepoints of blood sampling are described. Were blood samples also taken at -21, -18, -14 and -10dpc from animals that were challenged at 7dpv? Please clarify.
The text has been changed in lines 141-145 as follows: “Clotted blood from Combo-21 or O3039-21 groups were sampled on -21, -18, -14, -10, -7, -4, 0, 1–7, 11, 14, 18, 21, 25, 28 and 32 dpc and the animals, while Combo-7 or O3039-7 groups were sampled on -7, -4, 0, 1–7, 11, 14, 18, 21, 25, 28 and 32 dpc. The unvaccinated control animals were sampled on 0, 1–7, 11, 14, 18, 21, 25, 28 and 32 dpc.
- Line 196: Was the difference of clinical outcome statistically significant between vaccinated groups and unvaccinated group?
Yes, the text is amended as follows, “The difference in clinical outcome between the 4 vaccine groups was not significant (p=0.561; Fisher’s Exact Test, two-sided) but when compared with the unvaccinated control groups there was a significant difference (p=0.01129; Fisher’s Exact Test, two-sided).”
- Figure 1: Please correct the legend, because the dashed lines cannot be discriminated from the normal lines. Figure 3 has a bit of the same problem.
The lines have been coloured to differentiate the different groups.
- Figure 2: The boxplot is not clear. For some groups there is a grey box and for others not; for some groups there is a dashed line, while for others there are only diamonds visible. Shouldn't it all look the same? Please also explain in the figure's description how to read the figure.
A boxplot is a standardized way of displaying the distribution of data based on a five number summary (“minimum”, first quartile (Q1), median, third quartile (Q3), and “maximum”). The size of the box depends on the distribution of the data within the group. Therefore, some boxes are bigger than the others. If there is no box, this means within the group the values are the same. The boxes show the interquartile range (IQR) showing the middle 50% of scores (median; represented by the thick line) and the horizontal lines above and below the boxes are the maximum and minimum values for the group that is joined by the vertical line joining the IQR. A description is added to the figure now.
- Figure 4: Small table, difficult to read. Please include averages per group.
The orientation is now changed to landscape and fitted to a new page. We would request the Journal to allot a separate page for Figure 4.
- Line 254: replace O-AG with O-ALG
Changed.
- Line 260: IDL challenge is not introduced earlier. It should probably be introduced in line 129.
Added (presently line 133)
- Lines 312-316: It is not clear whether anti-pyretics and analgesics were used in this study. If so, please include the relevant information in the M&M section.
Added as suggested between lines 136-139 in the M&M section as follows: “Pain relief was given pre-emptively on days 0–4 after challenge and then on the onset of moderate clinical signs using Flunixin meglumine @3.33 mg/Kg body weight equivalent to 1 ml/15 Kg body weight (Finadyne pour-on 50 mg/ml; MSD Animal Health, the Netherlands).”
- Lines 317-319: Is it novel to use swabs for FMD diagnostics? If not, please refer to studies in which this has been done before.
Yes, this is an easy way of sampling for FMD. Many publications are available and is recommended for FMD surveillance. References are added in line 150 (citations 16-20).
- Lines 344-350: The conclusion does not address the aim of this study (lines 91-96). Especially the phrase "and make recommendations on vaccine strains to be included into vaccine banks". Therefore, the authors are requested to provide their recommendation. Should only O1 Manisa, only O3039 or the combo by included in vaccine banks?
Added in lines 376-379 as follows: “Given the epidemiological situation of FMD around the world and co-circulation of serotype O viruses belonging to different genotypes and lineages, we recommend that both O1 Manisa and O3039 are included in the vaccine banks.”
Reviewer 2 Report
This study is very practical to use available FMD vaccine to protect recent prevalent FMD virus. However, it will be better to revised bellow.
Discussion
Please discuss clearly why the antigen matching test by WRLP indication and your research result is different. This means that the antigen matching test by WRLP might not be useful to select candidate vaccine strain.
Author Response
This study is very practical to use available FMD vaccine to protect recent prevalent FMD virus. However, it will be better to revised bellow.
Discussion
Please discuss clearly why the antigen matching test by WRLP indication and your research result is different. This means that the antigen matching test by WRLP might not be useful to select candidate vaccine strain.
The following paragraph is added between lines 359-367.
Antigen matching studies by national and international reference laboratories give an indication of the match between new emerging isolates and the current vaccine strains. These studies are generally performed with reference sera raised using commercial vaccine formulations (payload equivalent to 3-6 PD50/dose) and not emergency vaccine formulations (payload that provide >6 PD50/dose). Whereas r1-values provide valuable indications of vaccine efficacy when assessing homologous systems, the interpretation is less clear when the vaccine and virus are heterologous. In vivo challenge studies such as the one described here present valuable data as to the efficacy of heterologous emergency vaccines.